# A Rapid Method for Screening Pathogen-Associated Molecular Pattern-Triggered Immunity-Intensifying Microbes

**DOI:** 10.3390/plants13162185

**Published:** 2024-08-07

**Authors:** Jing-Lin Zheng, Jia-Rong Li, Ai-Ting Li, Sin-Hua Li, Sabrina Diana Blanco, Si-Yan Chen, Yun-Ru Lai, Ming-Qiao Shi, Tsung-Chun Lin, Jiunn-Feng Su, Yi-Hsien Lin

**Affiliations:** 1Department of Plant Medicine, National Pingtung University of Science and Technology, Pingtung 912301, Taiwan; b10127018@gmail.com (J.-L.Z.); believe1206920@gmail.com (J.-R.L.); vivian12451245@gmail.com (A.-T.L.); nagihiko070421@gmail.com (S.-H.L.); sabrinablanco9779@gmail.com (S.D.B.); selina23464@gmail.com (S.-Y.C.); rita891206@gmail.com (Y.-R.L.); a0984436338@gmail.com (M.-Q.S.); 2Department of Tropical Agriculture and International Cooperation, National Pingtung University of Science and Technology, Pingtung 912301, Taiwan; 3Plant Pathology Division, Taiwan Agricultural Research Institute, Ministry of Agriculture, Taichung 413008, Taiwan; tclin@tari.gov.tw (T.-C.L.); forte9135101@tari.gov.tw (J.-F.S.)

**Keywords:** agricultural management, bacterial wilt, beneficial microbes, disease control, plant immunity

## Abstract

PAMP-triggered immunity (PTI) is the first layer of plant defense response that occurs on the plant plasma membrane. Recently, the application of a rhizobacterium, *Bacillus amyloliquefaciens* strain PMB05, has been demonstrated to enhance flg22_Pst_- or harpin-triggered PTI response such as callose deposition. This PTI intensification by PMB05 further contributes to plant disease resistance to different bacterial diseases. Under the demand for rapid and large-scale screening, it has become critical to establish a non-staining technology to identify microbial strains that can enhance PTI responses. Firstly, we confirmed that the expression of the *GSL5* gene, which is required for callose synthesis, can be enhanced by PMB05 during PTI activation triggered by flg22 or PopW (a harpin from *Ralstonia solanacearum*). The promoter region of the *GSL5* gene was further cloned and fused to the coding sequence of *gfp*. The constructed fragments were used to generate transgenic *Arabidopsis* plants through a plant transformation vector. The transgenic lines of At*GSL5*-GFP were obtained. The analysis was performed by infiltrating flg22_Pst_ or PopW in one homozygous line, and the results exhibited that the green fluorescent signals were observed until after 8 h. In addition, the PopW-induced fluorescent signal was significantly enhanced in the co-treatment with PMB05 at 4 h after inoculation. Furthermore, by using At*GSL5*-GFP to analyze 13 *Bacillus* spp. strains, the regulation of PopW-induced fluorescent signal was observed. And, the regulation of these fluorescent signals was similar to that performed by callose staining. More importantly, the *Bacillus* strains that enhance PopW-induced fluorescent signals would be more effective in reducing the occurrence of bacterial wilt. Taken together, the technique by using At*GSL5*-GFP would be a promising platform to screen plant immunity-intensifying microbes to control bacterial wilt.

## 1. Introduction

Under the threat of diseases to plant crop yield and quality, disease control methods that reduce the damage caused by pathogens are a very important topic to explore. Under this demand, in addition to utilizing fungicides to reduce the inoculums of pathogens in the field, improving the resistance of plants by alternative strategies is also a feasible idea. Among them, plant disease resistance can be enhanced by using the avirulent strains of specific pathogens, by transferring defense relative genes in transgenic plants, or by using beneficial microorganisms [1,2,3,4,5,6]. The application of beneficial microorganisms to improve plant disease resistance is commonly achieved by antagonistic activities, and by inducing plant resistance [7,8,9,10]. Recently, the beneficial microorganism, *Bacillus amyloliquefaciens* PMB05, has been demonstrated to possess qualities that can intensify plant immunity responses triggered by pathogen-associated molecular patterns (PAMPs), such as the flg22 (in flagellin) and PopW (a harpin) from plant pathogenic bacteria. This intensification of PAMP-triggered immunity (PTI) response only occurs in the co-existence of bacterial PAMPs and PMB05 [11,12,13]. Moreover, this effect would further provide resistance to different diseases, especially on the bacterial wilt of tomato and *Arabidopsis thaliana* [11,12,14,15,16,17,18]. Using bacterial strains that can increase plant defense signals during a pathogen invasion would be a novel way to improve disease resistance.

During the activation of PTI, callose deposition is not only a hallmark signal but it also plays an important role in plant disease resistance. During a pathogen invasion, callose is deposited between the cell membrane and the cell wall of plants. The thickening of the cell wall serves as a barrier to further prevent pathogen penetration, improving plant resistance against pathogens [19,20,21]. Some studies demonstrated that callose deposition was highly correlated to improving disease resistance and intensifying other PTI signals induced by plant ferredoxin-like protein (PFLP) and *B. amyloliquefaciens* PMB05 [12,13,22]. In the *GLUCAN SYNTHASE-LIKE* (*GSL*) gene family in *A. thaliana*, the *GSL5* gene is required for callose formation and deposition to improve penetration resistance to fungal pathogens [20]. Therefore, *GSL5* could be used as an index gene that can further be used to analyze the occurrence of PTI. With this in mind, a method for evaluating or screening PTI-regulating microbial strains through *GSL5* expression could be established. Green fluorescent proteins (GFPs) from jellyfish can be successfully used as reporter genes in many biological systems [23,24,25]. Therefore, transgenic plants that have established using *GFP* to show the level of *GSL* expression may have the potential to evaluate the strength of plant immunity.

In this study, we first evaluated the gene expression level of *GSL5* under the treatment with *B. amyloliquefaciens* PMB05 upon PAMP induction. Then, the upstream 1.4 Kb DNA fragment containing the promoter region of the *GSL5* coding sequence was cloned and fused to *GFP* as the reporter gene in a binary vector for plant transformation. Subsequently, *Agrobacterium*-mediated transformation was carried out to obtain At*GSL5*-GFP transgenic lines. After a homozygous line was obtained, its changes in green fluorescent signal were further evaluated at different time points after infiltrating with PAMPs. In addition, *B. amyloliquefaciens* PMB05 was used to establish the model on the regulation of fluorescent reaction in the presence of PAMPs. After that, the effects of other *Bacillus* spp. strain fluorescent reaction on At*GSL5*-GFP and callose deposition under the triggering of PTI were evaluated. Finally, the effects of all the *Bacillus* spp. strains on disease resistance to bacterial wilt were assayed. This study demonstrated that the At*GSL5*-GFP transgenic line could be used as a platform for screening plant immunity-intensifying biocontrol microbial strains against bacterial wilt.

## 2. Materials and Methods

### 2.1. Growth Conditions of Arabidopsis thaliana Plants and Bacteria

The *Arabidopsis thaliana* ecotype Columbia (Col-0) was used in this study. The seeds were sowed in sterile peat moss, and then the germinated seedlings were transplanted individually into new pots one week later. Four-week-old seedlings were used as the test materials for all the assays and *Agrobacterium*-mediated transformation. The seedlings were grown in a plant growth chamber (Hipoint, Kaohsiung, Taiwan) at 22 °C with 16 h of light and 8 h of darkness. The *Bacillus* spp. used in this study are the strains isolated from the rhizosphere of different plants. These strains were cultured on a nutrient agar (NA) plate and incubated at 28 °C for 24–48 h before use in the subsequent experiments. *Ralstonia solanacearum* Rd15, which exhibits a strong virulence of bacterial wilt on *A. thaliana* [26], was incubated on a 2,3,5-triphenyltetrazolium chloride (TZC) agar plate at 28 °C for 48 h. *Agrobacterium tumefaciens* GV3101 was grown in Luria–Bertani (LB) broth or on LB agar plates containing 50 μg/mL of gentamycin at 28 °C.

### 2.2. Analysis of GSL5 Gene Expression

To understand the changes in *GSL5* (glucan synthase-like 5) gene (AT4G03550) expression in *Arabidopsis thaliana* induced by PAMP, flg22_Pst_ was used for an analysis. The flg22_Pst_ was synthesized by LifeTien LCC (South Plainfield, NJ, USA) and dissolved in 25 mM Tris-HCl buffer (pH 7.5) to prepare a 1.0 μM of stock solution, and the final concentration was 0.5 μM in each treatment [13]. Four-week-old leaves were infiltrated with flg22_Pst_ mixed with 25 mM Tris-HCl buffer in equal volumes. To evaluate the effect of *Bacillus amyloliquefaciens* PMB05, the bacterial suspension adjusted to OD_600_ at 0.3 was used to mix with flg22_Pst_ in equal volumes before infiltration. The leaves were collected at 0, 4, and 12 h after infiltration. After isolating the total RNA, the cDNA was prepared for a quantitative real-time PCR [11]. The quantitative real-time PCR was performed with 200 ng of cDNA and 500 nM of each gene-specific primer in 1 × iQ™ SYBR Green supermix reagent (Bio-Rad, Hercules, CA, USA). Specific primers, gsl5-F (5′-CGGCAAAAGCTCACATACGG-3′) and gsl5-R (5′-CCCAGCCAGTTGGGATGAAT-3′), were synthesized for the quantitative real-time PCR. The reaction has two stages: Stage 1 was 95 °C for 3 min; Stage 2 was 95 °C for 10 s, and 50 °C for 30 s with 35 cycles. The reaction was carried out in a QIAquant 96 5plex (Qiagen, Hilden, Germany). The expression of *tubulin* was used as a reference gene [27], and the relative fold induction was normalized by the treatment at 0 h. At least five samples of each treatment were analyzed as repeats in this assay.

### 2.3. Construction of GSL5::GFP

To generate transgenic *Arabidopsis* plants, the DNA fragment containing *GSL5*::GFP needed to be constructed in a binary vector before plant transformation. Based on the sequence of *GSL5* (AT4G03550), specific primers upstream of the coding sequence GSL5PF and GSL5PR (Appendix A) were designed from the predicted promoter region upstream of the start codon (Figure 1). The 1410 bp of amplified fragments with 100% identity upstream of the coding sequence of *GSL5* were cloned using a pGM-T Cloning kit (GeneMark, Taichung, Taiwan) to obtain pGMT-*GSL5* in *Escherichia coli* DH5α. The specific primer for the *GFP* coding sequence (760 bp) was amplified by GFPPF and GFPPR2, and it was further cloned to obtain pGMT-GFP. The pGMT-*GSL5* was double digested with *Eco*RI and *Spe*I (Roche, Mannheim, Germany) and ligated with the digested pGMT-GFP to obtain pGMT-*GSL5*::GFP. To confirm the correctness, GSLPF and GFPPR2 were used for amplification to confirm the size of the DNA fragments before sequencing. The construction of a plant transformation vector was carried out as follows: after the pGMT-*GSL5*::GFP and pBI121were double digested with the restriction enzymes shown in Figure 1, the DNA fragments were filled by using Large (Klenow) Fragment (New England Biolabs, Ipswich, MA, USA) before blunt end ligation. After the pBI121 *GSL5*::GFP was obtained, it was confirmed by GSL5PF/PBI-IndR [26] and used for the amplification to confirm the size of the 2.4 Kb DNA fragments before sequencing. All the constructed plasmids were sequenced and compared using the Blast+ software (version 2.15.0) at the National Center for Biotechnology Information [28].

### 2.4. Plant Transformation and Transgenic Plant Screening

After the constructed plasmid pBI121-*GSL5*::GFP was transformed into *Agrobacterium tumegaciens* GV3101, the plant transformation was performed with the floral-dip method [29]. The transgenic seeds of At*GSL5*-GFP were screened by a rapid screening method [30] on a 1/2 Murashige and Skoogs medium containing 50 g/mL of kanamycin. The genomic DNA was isolated from the leaf of T_0_ transgenic lines and amplified with the primers *GSL5*PF and PBI-IndR to confirm the insertion of the transgene. Furthermore, T_1_ and T_2_ plants for each line were confirmed using the same method, and the homozygous T_3_ transgenic plants were used for the subsequent assay.

### 2.5. Leaf Fluorescence Image Analysis of *At*GSL5-GFP

To explore if the green fluorescence signal of At*GSL5*-GFP could be activated in the presence of PAMP, flg22_Pst_ and PopW were used in the assay. The PopW was expressed and purified based on our previous study [12]. The final concentrations of flg22_Pst_ and PopW for infiltration were 0.5 μM and 0.5 mg/mL, respectively. The treated leaf areas were observed at 0, 1, 4, 8, 12, and 16 h after infiltration. Similarly, in analyzing the impact of *Bacillus* spp. strains on the display of fluorescent signals, the co-infiltrations of PopW and the bacterial suspensions of each *Bacillus* sp. were performed. Each bacterial strain was incubated in a nutrient broth at 28 °C for 24 h, followed by centrifugation at 12,000× *g* for 10 min. The pellet was suspended in sterilized distilled water, and its OD_600_ was adjusted to 0.3. The bacterial suspension was mixed with the PopW solution in equal volumes to reach the final concentration of 0.5 mg/mL. The observation was carried out using a fluorescence microscope (Leica, Wetzlar, Germany). The light source filter wavelength of the microscope was Excitation 480 ± 30 nm and Emission 505–535 ± 40 nm. The fluorescent intensity emitted was the accumulation of GFP. All the images under the same size were quantified using the ImageJ software (version 1.52a) [31]. Three repeats of each treatment were analyzed as repeats in one individual experiment.

### 2.6. Observation of Callose Deposition

To observe the effect of *Bacillus* spp. strains on PopW-triggered callose deposition, the leaves of *Arabidopsis thaliana* Col-0 were infiltrated with the mixture containing PopW and the bacterial suspension of each *Bacillus* sp. strain. The infiltrated leaf samples were collected for callose staining as described in our previous study [12]. After the treated leaf strips were collected at 8 h post-infiltration, the samples were incubated and stained with 0.01% aniline blue (Sigma, St. Louis, MO, USA) in 0.1 M phosphate buffer (pH 8.0) for 2 h. The observation was carried out using a fluorescence microscope with an Excitation/Emission at 340–380/400–425 nm filter set (Leica, Wetzlar, Germany). The fluorescent intensity of callose deposition in each treatment was assayed by the ImageJ software under a consistent threshold [31]. Ten samples of individual leaves in each treatment were collected as repeats.

### 2.7. Disease Severity Assay

To evaluate the efficacy in reducing the disease severity of bacterial wilt by *Bacillus* spp. strains, the diseased soil method was utilized in this assay [12]. The bacterial cells of *Ralstonia solanacearum* Rd15 grown on a TZC agar plate were washed with sterilized distilled water to prepare the bacterial suspension with OD_600_ at 0.3. The bacterial suspension was mixed with a 10-fold volume of peat moss to prepare the diseased soil. Ten 3-week-old tomato plants were soaked in a bacterial suspension of each tested *Bacillus* sp. strain for 30 min, and then transplanted into the diseased soil two days later. The disease indexes of wilting symptoms on the *Arabidopsis* plants were rated from 0 to 4 (0: no wilting, 1: one leaf wilting, 2: two leaves wilting, 3: three leaves wilting, and 4: plant death), and then the total number (N = 10) of plants with different levels of wilting symptoms were counted. The disease severity of each treatment was calculated using the following formula: [(0 × N0 + 1 × N1 + 2 × N2 + 3 × N3 + 4 × N4)/(4 × N)] × 100% [32].

### 2.8. Statistical Analysis

Statistical analyses were performed using the SPSS Statistics software for Windows, version 25 (IBM Corp., Armonk, NY, USA). The analysis of variance (ANOVA) and post hoc tests (Dunnett’s T3) were used to analyze the significant differences between the treatments in the assays (*p* < 0.05). When comparing the differences between the two treatments, the *t*-test was used for the analysis.

## 3. Results

### 3.1. Gene Expression Changes in GSL5 Gene in Response to flg22_Pst_ Activation

To assay the *GSL5* gene expression, distinct treatments were carried out with *A. thaliana* Col-0 by infiltration and analyzed at 4 h and 12 h after treatment. The results of 4 h after treatment showed that flg22_Pst_ alone did not significantly increase *GSL5* expression compared to the blank treatment. However, the mixture of flg22_Pst_ and *B. amyloliquefaciens* PMB05 induced a 1.32-fold gene expression compared to the blank treatment (Figure 2). The results of 12 h after inoculation showed that the gene expression of both flg22_Pst_ alone and the mixture of flg22_Pst_ and PMB05 were significantly increased compared to the blank treatment. Compared to the blank treatment, the treatment of flg22_Pst_ alone could induce a 1.48-fold gene expression. Meanwhile, the treatment of the mixture could induce a 2.30-fold gene expression. The gene expression in the treatment of the mixture was higher than that of the flg22_Pst_-alone treatment (Figure 2).

### 3.2. Plasmid Construction for Plant Transformation

To construct a plasmid for plant transformation, it was necessary to perform the construction of placing the DNA fragment of the *GSL5* promoter fused with the GFP coding sequence into the binary vector pBI121. After pBI121 *GSL5*::GFP was obtained, it was further used for plant transformation.

### 3.3. Confirmation of *At*GSL5-GFP Transgenic Line

To evaluate the correctness of the transgenic plants, four kanamycin-resistant T_0_ plants were analyzed. The appearance of the four transgenic lines was not significantly different from that of non-transgenic plants (Figure 3A). Among them, a specific 2.2 kb amplicon was detected by using the genomic DNA from line number 1, 2, and 4 with *GSL5*PF/GFPPR2 (Figure 3B). The homozygous line of At*GSL5*-GFP, 1–13, was selected for further experiments. The T_1_ seeds of line number 1–13 showed a 93.4% germination rate while sown on a medium containing kanamycin. All the germinated T_2_ plants could grow green leaves normally (Figure 3C). The results of the PCR amplification with GFPPF/PBI-IndR showed that the 20 T_2_ plants tested could 100% carry the transgenic DNA fragment of approximately 1.0 kb amplicon (Figure 3D). The T_2_ plants from line number 1–13 were used for further analysis.

### 3.4. Leaf Fluorescence Performance *At*GSL5-GFP upon PAMP Activation

To evaluate if the expression of GFP could be activated upon PAMP induction, the leaves of At*GSL5*-GFP were infiltrated with flg22_Pst_ or PopW from *R. solanacearum*. The results exhibited that there was no signal in the blank treatment at all the time points. In the treatments with flg22_Pst_ or PopW, the fluorescent signals began to appear at 8 h after infiltration. And, the fluorescent signals were increased with time (Figure 4A). In terms of the quantitative results, the fluorescence intensity induced by PopW was significantly higher than that by flg22_Pst_ at all the time points. Comparing at 16 h after infiltration, the fluorescence intensity induced by PopW was 2.4-folds higher than that by flg22_Pst_ (Figure 4B).

### 3.5. Regulation of PopW-Induced GFP Fluorescence by Bacillus spp. Strains on *At*GSL5-GFP

To understand the regulation of GFP fluorescent signal by *Bacillus* spp. strain in the At*GSL5*-GFP plants, the assay was conducted with and without PopW induction. In the results without PopW induction, except for the strain PMBT03, all the other strains did not show GFP fluorescent signals. As a result of PopW induction, it can be observed that this protein can induce the production of weak fluorescent signals. The subsequent results exhibited that its co-treatment with eight *Bacillus* spp. Strains, including PMB05, PMBT01, PMBT02, PMBT12, PMBT17, PMBT21, PMBT33, and PMBT03, could enhance the fluorescent signal induced by PopW. However, five strains including PMB08, PMB09, PMB13, PMBT15, and PMB31 weakened the fluorescent signal induced by PopW (Figure 5A).

In the quantitative assay, the intensity in the PopW treatment with PMB05, PMBT01, PMBT02, PMBT12, PMBT17, PMBT21, PMBT33, and PMBT03 were significantly increased over 5.74-folds than that with PopW alone. However, the other strains were unable to significantly increase the fluorescence intensity induced by PopW. Among them, the treatment with strains such as PMB08, PMB09, PMB13, PMBT15, and PMB31 reduced the fluorescence intensity upon the PopW treatment significantly. In addition, PMBT11 and PMBT25 had no significant difference in the fluorescence signal intensity induced by PopW compared with the control treatment with PopW alone (Figure 5B).

### 3.6. Regulation of PopW-Mediated Callose Deposition by Bacillus spp. Strains

The regulation of PopW-mediated callose deposition by distinct *Bacillus* spp. strains was observed at 8 h post-infiltration. The results revealed that the fluorescent callose signals were induced by PopW in *A. thaliana*. However, only PMBT03 induced a weak fluorescent signal of callose. Fluorescent signals could not be observed in the other tested strains. The subsequent results exhibited that its co-treatment with eight *Bacillus* spp. strains, including PMB05, PMBT01, PMBT02, PMBT12, PMBT17, PMBT21, PMBT33, and PMBT03, could produce a more abundant fluorescent signal induced by PopW. However, five strains including PMB08, PMB09, PMB13, PMBT15, and PMB31 produced less fluorescent signal induced by PopW (Figure 6A).

In terms of the quantitative results, the callose signals were stronger in the treatment with PMB05, PMBT01, PMBT02, PMBT12, PMBT17, PMBT21, PMBT33, and PMBT03 than in PopW alone. On the other hand, the callose signals in the treatment with PMB08, PMB09, PMB13, PMBT11, PMBT15, PMBT25, and PMB31 did not have a significant difference with that in PopW alone (Figure 6B).

### 3.7. Disease Resistance Affected by Bacillus spp. Strains

To evaluate whether the disease resistance was affected by *Bacillus* spp. strains, the disease severity of bacterial wilt was assayed. Aiming at the regulation of plant immune response, we selected 4, 3, and 2 strains from three categories of improvement, reduction, and no effect, respectively, for this assay. The results revealed that the disease severity following PMB05 (16.3%, 36.6%), PMBT03 (24.3%, 43.3%), PMBT21 (22.7%, 35.3%), and PMBT33 (25.7%, 37.3%) were lower than control treatment (46.3%, 88.0%) at 2- and 4-weeks post-inoculation, respectively. The disease severity following PMB08 (78.3%, 100.0%), PMB09 (73.0%, 99.0%), and PMB13 (77.0%, 97.0%) were higher than control treatment at 2- and 4-weeks post-inoculation, respectively (Figure 7A). Two *Bacillus* spp. strains, PMBT11 and PMBT15, which had no effect on the plant immune response induced by PopW, had no significant effects on the severity and symptoms of bacterial wilt. In terms of the wilt symptoms that appear at 4-week post-inoculation, most plants treated with PMB05, PMBT03, PMBT21, and PMBT33 were weak or symptomless (Figure 7B).

## 4. Discussion

Protecting crops against the occurrence of plant diseases is an important issue in agricultural science. The use of antagonistic microorganisms in the control of plant diseases not only increases crop production, but also ensures the safety of agricultural products. Under this demand, it is important to screen antagonistic microorganisms for controlling plant diseases. In the screening of antagonistic microorganisms, the differences in their functions against plant pathogens can be targeted. Besides the screening method for antagonistic effects on pathogenic bacteria [33,34,35], according to the current understanding, there is still no method that can quickly and effectively screen microorganisms that can induce disease resistance or intensify plant immunity in plants. The purpose of this study was to develop a screening platform that can screen microorganisms that have the ability to intensify plant immune response. The triggering of plant immunity can be mainly divided into two layers of recognition. The first layer of defense response is activated by the recognition of molecules (pathogen-associated molecular pattern, PAMP) produced by pathogens on the surface of plant cells. This defense response is called PAMP-triggered immunity (PTI). The second layer of defense response is triggered by the recognition of effector protein, which is called the effector-triggered immunity (ETI). Since PTI occurs before or at the first moment of contact between the pathogen and plant cells, it is often regarded as the first line of defense [36]. In our previous studies, we have proven that a microbial strain isolated from the soil, *B. amyloliquefaciens* PMB05, can effectively intensify the PTI signals to contribute excellent control effects on a variety of plant diseases [11,12,16,17,18,37]. Therefore, how to develop a platform that can screen such microorganisms has become the key to the development of next-generation novel microorganisms for disease control. *B. amyloliquefaciens* PMB05 can also be used as a model strain to develop a microbial screening platform for improving plant immunity.

In response to the demand of establishing a screening platform through transgenic plants, it is feasible to select indicator genes related to plant immune responses. Since callose is one of the indicator signals of plant immunity, this reaction can prevent the invasion of pathogens [38]. At the same time, during the plant immune response enhanced by *Bacillus amyloliquefaciens* PMB05, a large amount of callose deposition can be observed whether fungal or bacterial cells are used as elicitors [12,18]. Therefore, we believe that the callose synthetic gene can be used as an indicator for screening microorganisms that regulate plant immunity. In the synthesis of callose, 12 *GLUCAN SYNTHASE-LIKE* (*GSL*) genes have been identified in *Arabidopsis thaliana*. Among them, the gene proven to be related to disease resistance to fungal diseases is *GSL5* (*POWDERY MILDEW RESISTANT4*, *PMR4*) [20,38,39]. In this study, we demonstrated that the expression of At*GSL5* in *Arabidopsis* was induced by flg22, and its expression level was further intensified by the treatment of *B. amyloliquefaciens* PMB05. These results demonstrated that *B. amyloliquefaciens* PMB05 intensified the callose deposition in PTI, which was consistent with the trend in *GSL5* gene expression.

After constructing a reporter gene using the promoter region of the *GSL5* gene combined with the coding region of green fluorescent protein, *Arabidopsis* transgenic plants were generated by *Agrobacterium*-mediated transformation. During the growth of the At*GSL5*-GFP transgenic plants obtained after screening, it was observed that the appearance of the transgenic lines was not significantly different from that of the wild-type Col-0 plants. Among them, the seeds of lines 1–13 obtained after self-pollination not only showed kanamycin resistance on the 1/2 MS medium after germination, but were also 100% detectable on transgenic DNA fragments in PCR detection. It was speculated that the line 1–13 of At*GSL5*-GFP was a homozygous line and could be used for subsequent experiments. This transgenic line was used to observe its performance on green fluorescence regulated by different microbial strains upon PAMP induction. We first observed that the fluorescent signals were induced by distinct bacterial PAMPs in this transgenic line, and more fluorescent signals appeared as time went by. This trend is consistent with the results of gene expression. Among them, the fluorescence intensity induced by PopW was significantly higher than that of the flg22_Pst_ treatment. This result is also consistent with our previous study that proved that treatment with harpin can provide a more stable and stronger plant immune signal [13]. These results indicate that At*GSL5*-GFP could be subsequently used to assay the regulation of plant immunity by microbial strains upon PopW treatment. According to the regulation of the PopW-induced fluorescence signal by microbial strains, microbial strains can be divided into three categories: increased immune signal, decreased immune signal, and no response. Further results showed that the increase or decrease in the fluorescence signal regulated by the microbial strains indeed exhibited the same trend as the callose deposition. It was speculated from the above results that six strains can increase fluorescence intensity, including *B. amyloliquefaciens* PMB05, which may enhance plant disease resistance, while five strains that reduce fluorescence intensity, including *Bacillus* sp. PMB08, may reduce plant disease resistance.

The inferred trend in plant immune regulation by *Bacillus* spp. strains obtained using At*GSL5*-GFP was consistent with the results obtained by the subsequent callose deposition analysis on the *A. thaliana* Col-0 plant. These results suggested that At*GSL5*-GFP can be used to analyze the regulation of plant immunity by *Bacillus* spp. strains. However, the *Bacillus* spp. strains that can reduce PopW-induced GFP fluorescence in At*GSL5*-GFP cannot actually inhibit PopW-induced callose deposition. It is speculated that analysis using At*GSL5*-GFP can more specifically analyze the regulation of plant immune responses by microbial strains than using callose deposition assay. In order to avoid being unable to judge the regulatory status due to the excessive fluorescence signal induced by PAMP on At*GSL5*-GFP after the addition of *Bacillus* spp. strain, 12 h after infiltration was selected as the time point for further analysis.

To confirm this speculation, nine strains were selected to analyze their control effects on bacterial wilt. The results showed that the PTI-intensifying strains (PMB05, PMBT03, PMBT21, and PMBT33) can significantly reduce the severity of bacterial wilt. On the other hand, the PTI-reducing strains (PMB08, PMB09, and PMB13) significantly made the occurrence of bacterial wilt faster and more severe. This result proves that the regulation of plant immune responses by soil microbial strains can become a key factor in the successful control of bacterial wilt disease using beneficial microorganisms. In addition, many reports have shown that rhizosphere bacterial strains with good antagonistic capabilities against *Ralstonia solanacearum* can also control plant diseases [7,12,40]. From this, it can be speculated that microbial strains with intensified plant immune responses may be more effective in controlling plant diseases if they also have antagonistic properties against pathogens. Compared with the need for microorganisms to produce antagonistic compounds against specific plant pathogens, it can be speculated that the use of microbial strains that intensify plant immunity may increase the spectrum of disease resistance. Although *B. amyloliquefaciens* PMB05 has been used to prove that the intensification of plant immune response can improve resistance to different diseases in a variety of crops, the evidence of the immune response regulation and control effects of microbial strains on different crops are important research directions before the application of these microorganisms.

Taken together, the above results indicate that the At*GSL5*-GFP transgenic *Arabidopsis* established in this study can evaluate the regulation of plant immune response by microorganisms in the presence of PopW. Screening microbial strains to control bacterial wilt in this way would be feasible and promising to identify microbial strains that possess the ability to control disease effects caused by distinct plant diseases.

## Figures and Tables

**Figure 1 plants-13-02185-f001:**
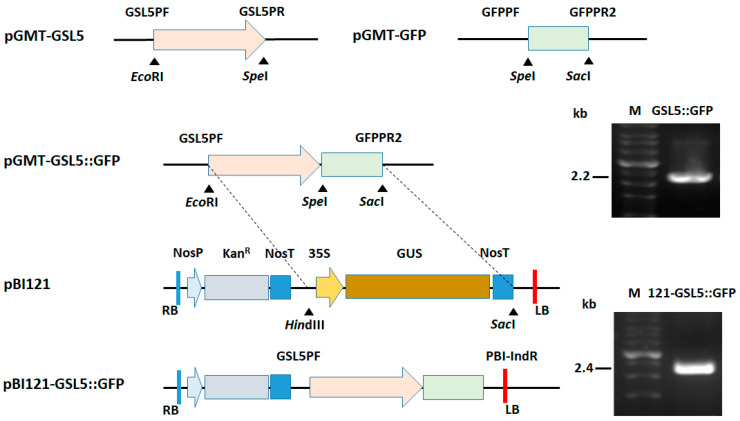
Construction and confirmation of transgenic *Arabidopsis thaliana* lines. The pGMT-*GSL5* and pGMT-GFP were two plasmids containing the amplified *GSL5* promoter region and *gfp* coding sequence, respectively, in the pGMT-vector. The pGMT-*GSL5*::GFP was obtained by the insertion of the *GSL5* promoter region to the pGMT-GFP. The DNA fragment from *Eco*RI to *Sac*I in the pGMT-*GSL5*::GFP was used to replace that from *Hin*dIII to *Sac*I in pBI121 to obtain pBI121-*GSL5*::GFP. Electrophoresis was carried out to confirm the correct size of the constructed fragments by PCR amplification with specific primers. Marked above and below the sequence are the corresponding primers and restriction enzyme sites, respectively.

**Figure 2 plants-13-02185-f002:**
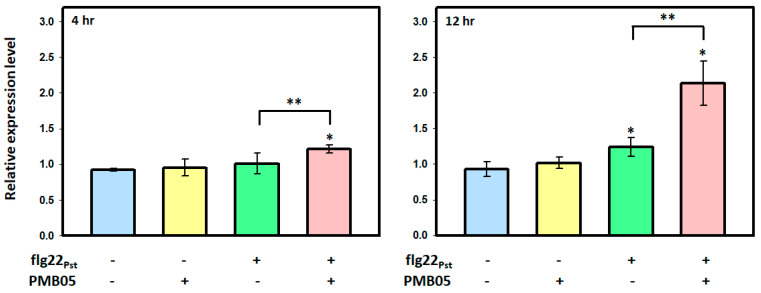
Effect of *Bacillus amyloliquefaciens* PMB05 on the *GSL5* gene expression upon the flg22_Pst_ treatment in *Arabidopsis thaliana* Col-0. Four-week-old leaves were infiltrated with flg22_Pst_ or a mixture of flg22_Pst_ and *B. amyloliquefaciens* PMB05. The infiltrated leaves were collected at 4, and 12 h post-infiltration to analyze gene expression. The fold induction of each treatment was normalized by mock treatment at 0 h. An asterisk indicates significant differences compared to the mock treatment based on a *t-*test (*p* < 0.05). A double asterisk indicates significant differences compared to the flg22_Pst_ treatment based on a *t*-test (*p* < 0.05).

**Figure 3 plants-13-02185-f003:**
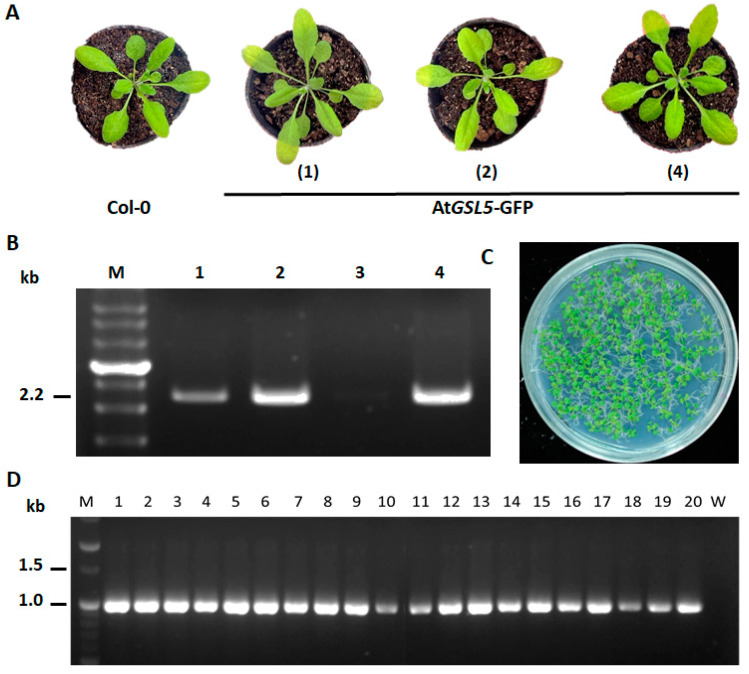
Screening of the transgenic *Arabidopsis thaliana* At*GSL5*-GFP homozygous line. (**A**) shows the seedings of the possible transgenic *A. thaliana* At*GSL5*-GFP lines (#1, #2, and #4) obtained from the seedlings grown on 1/2 MS medium with kanamycin. (**B**) indicates the specific amplicon amplified from the genomic DNA of At*GSL5*-GFP lines. (**C**) shows the seedlings of the possible homozygous At*GSL5*-GFP line 1–13 grown on 1/2 MS medium with kanamycin. (**D**) indicates the homozygous At*GSL5*-GFP line 1–13 confirmed by the PCR amplification. The PCR amplification was performed with specific primer GFPPF/PBI-IndR to confirm the DNA fragment. M indicates the DNA ladders (Genemark, Taichung, Taiwan).

**Figure 4 plants-13-02185-f004:**
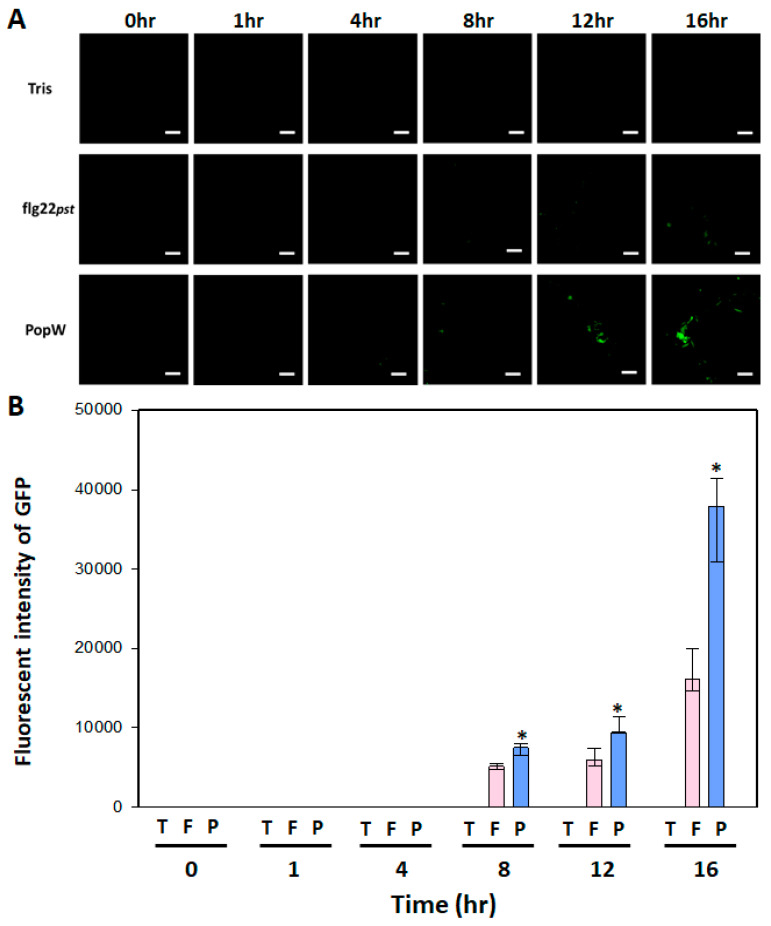
The PAMP-induced fluorescence signal performance of the transgenic At*GSL5*-GFP plant at different time points. The leaves of At*GSL5*-GFP were infiltrated with flg22_Pst_ or PopW and observed under fluorescent microscopy at distinct time points. (**A**) indicates the images of GFP fluorescence elicited by different treatments. The bar indicates 20 μm in length. (**B**) indicates the quantitative fluorescent intensity of GFP measured by Image J. The asterisks indicate significant differences between two PAMP treatments based on a *t*-test (*p* < 0.05).

**Figure 5 plants-13-02185-f005:**
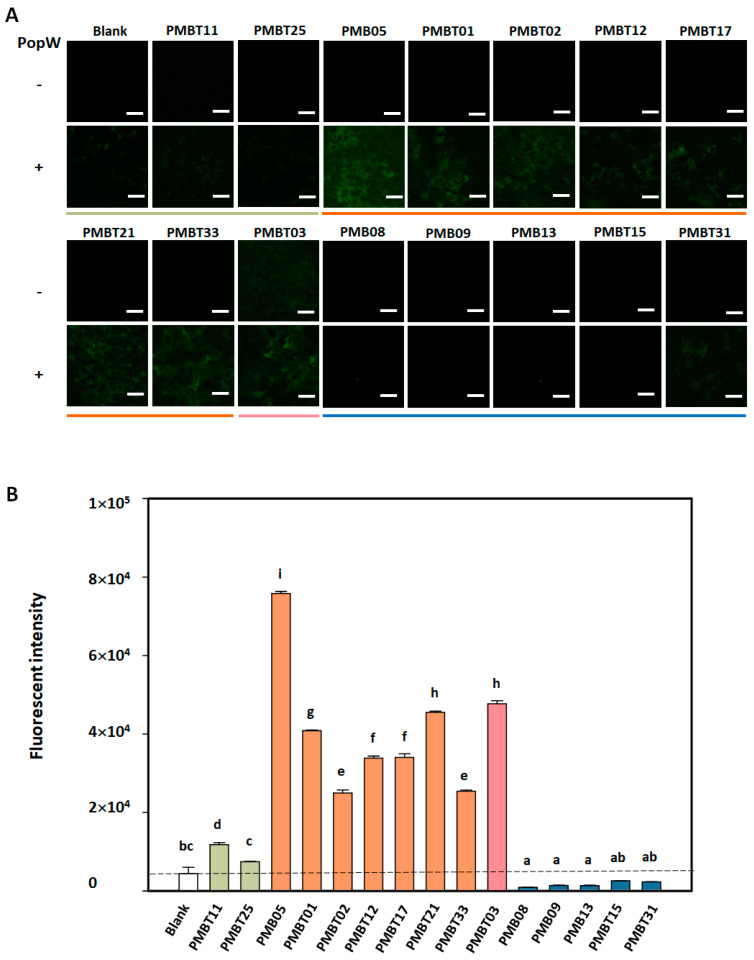
PopW-induced fluorescence signal regulated by *Bacillus* spp. strains on the leaves of the At*GSL5*-GFP plants. The assay was performed with the mixtures of PopW and bacterial suspensions of distinct *Bacillus* spp. strains at 0.5 mg/mL and 10^8^ CFU/mL, respectively. (**A**) reveals the image of the fluorescent signal of GFP regulated by *Bacillus* spp. strains upon PopW induction at 12 h post-infiltration. Blank indicates the treatment with Tris-HCl alone. The symbols “+” and “−” indicate the inclusion and exclusion of PopW, respectively. (**B**) reveals the quantitative PopW-induced fluorescent signal in distinct *Bacillus* spp. strain treatments as calculated by ImageJ. Ten infiltrated samples were used in each treatment as repeats. Different letters above the columns indicate significant differences between the distinct *Bacillus* sp. strains based on Tukey’s HSD test (*p* < 0.05).

**Figure 6 plants-13-02185-f006:**
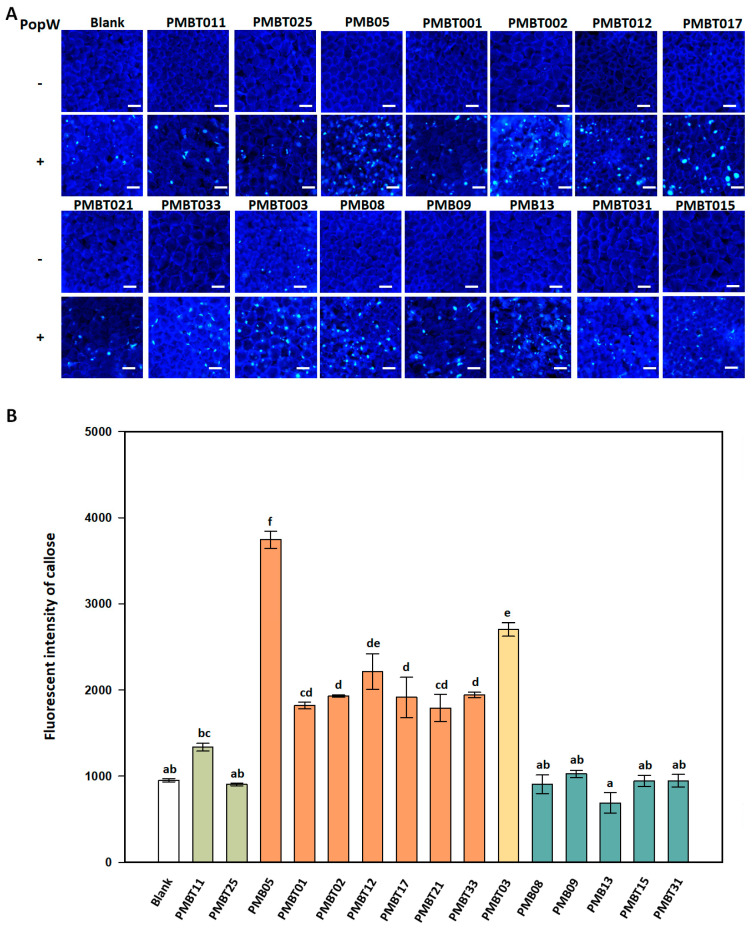
Effects of the *Bacillus* spp. strains on PopW-induced callose deposition in *Arabidopsis thaliana* Col-0. The assay was performed with the mixtures of PopW and bacterial suspensions of distinct *Bacillus* spp. strains at 0.5 mg/mL and 10^8^ CFU/mL, respectively. (**A**) indicates the images of callose deposition regulated by the *Bacillus* spp. strains upon the PopW treatment in the *A. thaliana* Col-0 plant. To observe the callose deposition, the infiltrated leaves were collected and stained with 0.01% aniline blue at 8 h post-infiltration. The scale bars indicate 50 μm in length. Blank indicates the treatment with Tris-HCl alone. The symbols “+” and “−” indicate the inclusion and exclusion of PopW, respectively. (**B**) indicates the quantitative fluorescent intensities of the PopW-induced callose deposition in the distinct *Bacillus* spp. strain treatments as calculated by ImageJ. Different letters above the columns indicate significant differences according to Tukey’s HSD test (*p* < 0.05).

**Figure 7 plants-13-02185-f007:**
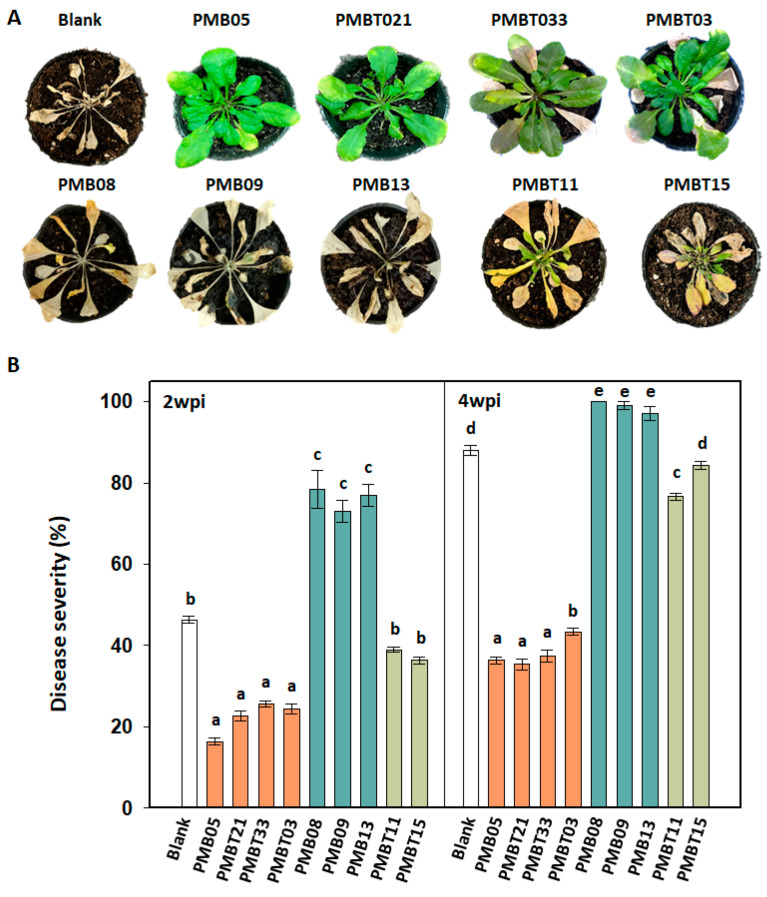
Effect of *Bacillus* spp. strains on the control of bacterial wilt in *Arabidopsis thaliana*. (**A**,**B**) exhibit the symptom appearance and disease severity of bacterial wilt post-inoculation, respectively. The treatments of *Bacillus* spp. strains were carried out by soaking the seedlings in the bacterial suspensions two days before inoculation. Blank indicates the treatment with sterilized water only. Then, the plants were transplanted into diseased soil containing *Ralstonia solanacearum* Rd15 as an inoculation to evaluate the occurrence of wilting symptoms at 2- and 4-week post-inoculation (wpi). Different letters above the columns indicate significant differences based on Tukey’s HSD test (*p* < 0.05).

## Data Availability

Data are contained within the article and Appendix A.

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
