# Peer review of "A Rapid Method for Screening Pathogen-Associated Molecular Pattern-Triggered Immunity-Intensifying Microbes"

_plants, 2024, doi:10.3390/plants13162185_

Round 1

Reviewer 1 Report

Comments and Suggestions for Authors

The manuscript submitted by Zhang et al. reported a rapid method for screening PTI-intensifying microbes using a GSL5pro-GFP reporter system. Based on Bacillus amyloliquefaciens PMB05 enhances flg22 or harpin-triggered PTI response such as callose deposition, the author transformed binary vector carrying AtGSL5pro drived GFP into Arabidopsis and obtained stable transgenic lines. The PTI-enhancing by PMB05 was confirmed on the transgenic lines with AtGSL5pro-GFP reporter system. Furthermore, a series of Bacillus strains were screened by this reporter system and found other three strains PMBT03, PMBT21 and PMBT33 reducing the disease severity of bacterial wilt. Generally, the paper is easy readable and provides a new simple strategy for screening intensifying microbes. I recommend to accept the manuscript after minor revision.

1.     L82: how long is the promoter region?

2.     L103-107: 28 ℃

3.     L112-113: give the used final concentration directly

4.     L118: formalize the writing of “1  iQ™ SYBR”

5.     L131: ‘amplified fragments’,length and detail location in the genome or the detail sequence

6.     L197-199: ‘disease indexes’, any reference for the grade standard of disease symptom caused by R. solanacearum

7.     Fig. 2, In method, the author mentioned gene expression of GSL5 were measured at 0, 1, 4, 8, 12, and 16 hrs after infiltration. why not the author show the whole results at different timepoints via a line chart?

8.     L232-242(section 3.2): most part of this section had mentioned in Method and showed the corresponding progress in Fig. 1. Try to plus the detail information including fragment length and start-stop in genome into Method and remove this section directly. This also eliminates the confusion of Fig. 1 results appearing after Fig. 2.

9.     Capital or small? Induced, Strains in L282; Mediated, Strains in L310;

10.   Italic spp. in L349

Author Response

The manuscript submitted by Zhang et al. reported a rapid method for screening PTI-intensifying microbes using a GSL5pro-GFP reporter system. Based on Bacillus amyloliquefaciens PMB05 enhances flg22 or harpin-triggered PTI response such as callose deposition, the author transformed binary vector carrying AtGSL5pro drived GFP into Arabidopsis and obtained stable transgenic lines. The PTI-enhancing by PMB05 was confirmed on the transgenic lines with AtGSL5pro-GFP reporter system. Furthermore, a series of Bacillus strains were screened by this reporter system and found other three strains PMBT03, PMBT21 and PMBT33 reducing the disease severity of bacterial wilt. Generally, the paper is easy readable and provides a new simple strategy for screening intensifying microbes. I recommend to accept the manuscript after minor revision.

Response:

Thanks to the reviewer’s contribution and comments, we have also revised the manuscript according to the suggestions.

  1. L82: how long is the promoter region?

Response:

Thanks for reminding. We revised this sentence to " Then, the upstream 1.4 Kb DNA fragment containing the promoter region of GSL5 coding sequence was cloned and fused to GFP as the reporter gene in a binary vector for plant transformation." based on the suggestion.

  1. L103-107: 28 â„ƒ

Response:

Thanks for reminding. Errors caused by symbols in file conversion have been corrected in manuscript.

  1. L112-113: give the used final concentration directly

Response:

Thank you. This important information has been added to the revised manuscript.

  1. L118: formalize the writing of “1  iQ™ SYBR”

Response

Thank you. The symbol was revised in the manuscript.

  1. L131: ‘amplified fragments’,length and detail location in the genome or the detail sequence

Response

Thank you. The information was added in the manuscript.

  1. L197-199: ‘disease indexes’, any reference for the grade standard of disease symptom caused by  solanacearum

Response

Thank you. The reference 32 was the original paper to scale the symptom o bacterial wilt of Ralstonia solanacearum (same as Pseudomonas solanacearum).

  1. 2, In method, the author mentioned gene expression of GSL5 were measured at 0, 1, 4, 8, 12, and 16 hrs after infiltration. why not the author show the whole results at different timepoints via a line chart?

Response:

Thank you. The gene expression obtained by using real-time PCR only determined at three time points post-infiltration: 0, 4, and 12 hours; for the presentation of green fluorescence in AtGSL5-GFP, we observed at five time points post-infiltration: 0, 1, 4, 8, and 12 hours. Therefore, only the results at 4 hours and 12 hours post-infiltration were presented in the gene expression analysis (Figure 2).

  1. L232-242(section 3.2): most part of this section had mentioned in Method and showed the corresponding progress in Fig. 1. Try to plus the detail information including fragment length and start-stop in genome into Method and remove this section directly. This also eliminates the confusion of Fig. 1 results appearing after Fig. 2.

Response:

Thanks for this valuable advice. We have removed redundant part in section 3.2 and consolidated the information into Materials and Methods (section 2.3). We believe the changes would reduce the confusion.

  1. Capital or small? Induced, Strains in L282; Mediated, Strains in L310;

Response:

Thank you. All of the typo was corrected.

  1. Italic spp. in L349

Response:

Thank you. I revised it accordling to the suggestion.

Reviewer 2 Report

Comments and Suggestions for Authors

In this study, the authors developed the system to screen the microbes to control bacterial wilt using PAMP responsive promoter of GSL5 gene. GFP analysis results were consistent with callose deposition and bacterial wilt results, indicating that PGSL5-GFP transgenic plants can be applied for screening the bacterial strains. Overall, the experiments were designed well and the results were convincing. I have several comments for the publication.

1. More background information about PAMPs (flg22 and PopW) used in this research is necessary at Introduction. 

2. In Figure 2, Fig. 2A and 2B were not designated.

3. In Figure 4, the authors investigated the GFP signal intensity with time course, however, the author showed the results at one time point in the Fig 5 and 6. Did you test them at different time points and choose the best time point? If so, you have to describe it in the result. If not, explain the reason. 

4. Total 15 strains were analyzed in GFP and callose deposition experiments (Fig. 5 and 6). However, 9 strains were tested in bacterial wilt assay. Explain this in the result.

5. More detailed information about the promoter of GSL5 is necessary in Fig. 1. The length of the promoter and the genomic location of the promoter. Does the promoter include the 5' UTR of GSL5?

Comments on the Quality of English Language

Minor editing of English language required

Author Response

In this study, the authors developed the system to screen the microbes to control bacterial wilt using PAMP responsive promoter of GSL5 gene. GFP analysis results were consistent with callose deposition and bacterial wilt results, indicating that PGSL5-GFP transgenic plants can be applied for screening the bacterial strains. Overall, the experiments were designed well and the results were convincing. I have several comments for the publication.

Response:

Thank you for your excellent suggestions. We have tried our best to revise the manuscript to make it clearer.

  1. More background information about PAMPs (flg22 and PopW) used in this research is necessary at Introduction. 

Responses:

Thank you for your suggestion. Because these PAMPs are small molecules derived from plant pathogenic bacteria, minor revisions have been made to the description.

  1. In Figure 2, Fig. 2A and 2B were not designated.

Response:

Thanks. Because Figure 2 is not divided into A and B, we marked the analysis time in the upper left corner of each picture. For errors we described in the results, we have corrected them in the manuscript.

  1. In Figure 4, the authors investigated the GFP signal intensity with time course, however, the author showed the results at one time point in the Fig 5 and 6. Did you test them at different time points and choose the best time point? If so, you have to describe it in the result. If not, explain the reason. 

Response:

Thank you! This is great advice. After careful consideration, we decided to explain it in the discussion (L437-440).

  1. Total 15 strains were analyzed in GFP and callose deposition experiments (Fig. 5 and 6). However, 9 strains were tested in bacterial wilt assay. Explain this in the result.

Response:

Thank you! This is a good suggestion and has been modified in the results.

  1. More detailed information about the promoter of GSL5 is necessary in Fig. 1. The length of the promoter and the genomic location of the promoter. Does the promoter include the 5' UTR of GSL5?

Response:

Thanks for reminding. We added the information based on the suggestion in materials and methods. In our experiment, because the sequence of more than 1 KB upstream the coding sequence of GSL5 has been cloned, a more detailed genetic analysis of the promoter sequence was not performed.